# Similarities and Differences of Hsp70, hsc70, Grp78 and Mortalin as Cancer Biomarkers and Drug Targets

**DOI:** 10.3390/cells10112996

**Published:** 2021-11-03

**Authors:** Rajani Rai, Amy L. Kennedy, Zitha Redempta Isingizwe, Pouya Javadian, Doris Mangiaracina Benbrook

**Affiliations:** 1Gynecologic Oncology Section, Obstetrics and Gynecology Department, Stephenson Cancer Center, University of Oklahoma Health Sciences Center, Oklahoma City, OK 73104, USA; Rajani-Rai@ouhsc.edu (R.R.); Pouya-Javadian@ouhsc.edu (P.J.); 2Pathology Department, University of Oklahoma Health Sciences Center, Oklahoma City, OK 73104, USA; Amy-Bosley@ouhsc.edu; 3Pharmaceutical Sciences Department, Stephenson Cancer Center, University of Oklahoma Health Sciences Center, Oklahoma City, OK 73104, USA; Zitha-Isingizwe@ouhsc.edu

**Keywords:** Hsp70, hsc70, Grp78, mortalin, chaperone, cancer, cellular localization, biomarker, prevention, combination therapy

## Abstract

**Background**: Upregulation of Heath Shock Protein 70 (HSP70) chaperones supports cancer cell survival. Their high homology causes a challenge to differentiate them in experimental or prevention and treatment strategies. The objective of this investigation was to determine similarities and differences of Hsp70, hsc70, Grp78 and Mortalin members of the HSP70 family encoded by *HSPA1*, *HSPA8*, *HSPA5* and *HSPA9* genes, respectively. **Methods:** Literature reviews were conducted using *HSPA1*, *HSPA5*, *HSPA8* and *HSPA9* gene or protein names or synonyms combined with biological or cancer-relevant terms. Ingenuity Pathway Analysis was used to identify and compare profiles of proteins that directly bind individual chaperones and their associated pathways. TCGA data was probed to identify associations of hsc70 with cancer patient survival. ClinicalTrials.gov was used to identify HSP70 family studies. **Results:** The chaperones have similar protein folding functions. Their different cellular effects are determined by co-chaperones and client proteins combined with their intra- and extra-cellular localizations. Their upregulation is associated with worse patient prognosis in multiple cancers and can stimulate tumor immune responses or drug resistance. Their inhibition selectively kills cancer over healthy cells. **Conclusions:** Differences in Hsp70, hsc70, Grp78 and mortalin provide opportunities to calibrate HSP70 inhibitors for individual cancers and combination therapies.

## 1. Introduction

Heat shock proteins (HSPs) play critical roles in cancer development, progression and drug resistance by assuring proper folding or degradation of unfolded or misfolded proteins (client proteins) [1,2]. Proper folding is critical for the function of proteins, and misfolded proteins can be toxic to the cell. Stressful conditions experienced by cancer cells often lead to increased misfolding of proteins and the need for molecular chaperones to maintain homeostasis. Thus, it is not surprising that HSP proteins are found to be elevated in cancer and targeted in the development of cancer treatment and prevention [3,4,5]. The HSPs function as complexes of chaperones, co-chaperones and nucleotide exchange factors (NEFs), in which each component plays a specific role [6]. In the human genome there are 332 genes encoding molecular chaperone and co-chaperones, and the HSP subset of these genes are subcategorized based on the molecular weights of their encoded proteins [6,7]. Upregulation of the 70 kD subset of HSP proteins (HSP70s) has been shown to be vital to the survival of cancer cells and has been targeted by multiple therapeutic strategies [5]. These HSP70 proteins are encoded by the *HSPA* genes and have been called by a variety of names in the scientific literature (Appendix A). Their similarity in size and homology have resulted in mislabeling, misinformation and misunderstandings about the specific identify of the individual HSP70 protein described [7]. For instance, the term Hsp70 has been used to refer to both Hsp70 encoded by *HSPA1* and heat shock cognate 70 (hsc70) encoded by *HSPA8* or to all 70 kD HSPs. The Hsp70 family member encoded by the *HSPA1* gene is the most well-characterized and targeted by therapeutic strategies in cancer [8,9]. Most of the recent reviews describing HSP70 proteins do not differentiate between the individual proteins [5,10,11,12,13].

The purpose of this analysis of literature and data is to evaluate how a subset of HSPA proteins are similar and different from the well-characterized Hsp70 protein. Three HSPA proteins were chosen for this comparison, hsc70, glucose regulated protein 78 (Grp78 encoded by *HSPA5*) and mortalin (encoded by *HSPA9*), because they are bound, and their complexes disrupted, by the cancer new investigational agent, Sulfur Heteroarotinoid A2 (SHetA2, NSC 726189) [14]. The natural functions of these proteins will be contrasted with their roles in cancer, the profiles of their protein binding partners will be compared, and their utilization as cancer biomarkers and targets for development of cancer prevention and treatment strategies will be reviewed.

## 2. Materials and Methods

The strategic approach of this investigation is illustrated in Figure 1.

### 2.1. Literature Review

Each of the authors independently performed literature searches in online databases (PubMed and Ovid MEDLINE) for articles related to Hsp70, hsc70, Grp78 and mortalin between 1965 and 2021. The search terms used were the *HSPA1A*, *HSPA5*, *HSPA8* and *HSPA9* gene and their synonyms listed in Appendix A in combination with neoplasia, cancer, immun*, exosome, membrane, nucleus, mitochondria, endoplasmic reticulum, blood, biomarker, drug, prevention, therapeutic, clinical trial, prognosis, or survival. Publications between 2019 and 2021 were first reviewed to establish themes and then earlier articles relevant to the themes were included to assure thoroughness of the review. The reference lists of the articles identified were examined for potential additional pertinent references. For specific findings that had numerous publications, either the original or the most recent publications were cited depending on the context of the sentence.

### 2.2. Bioinformatic Analysis

The synonyms and locations for the individual proteins were collated from GeneCards (https://www.genecards.org/ last accessed on 24 September 2021), UniPort (https://www.uniprot.org/ last accessed on 24 September 2021), National Center for Biotechnology Information (NCBI) (https://www.ncbi.nlm.nih.gov/ last accessed on 09/24/2021), and Ingenuity (Qiagen) databases. The proteins which bind to hsp70/*HSPA1*, Grp78/*HSPA5*, hsc70/*HSPA8* and mortalin/*HSPA9* were identified using the Ingenuity Pathway Analysis (IPA) Software (Qiagen) by inserting these genes into a newly created IPA pathway and using the “grow” tool with limits set to experimentally observed, direct protein–protein interactions. Lists of the identified proteins were compared to create profiles of proteins that uniquely bind to individual *HSPA*-encoded proteins and those that are bound by multiple *HSPA*-encoded proteins. The molecular, cellular, physiological and disease pathways of the profiles were identified by IPA using direct interactions only. The Cancer Genome Atlas (TCGA) data was probed for significant associations of tumoral hsc70 mRNA expression with patient survival probability using the University of Alabama Cancer (UALCAN) website (ualcan.path.uab.edu). The clinicaltrials.gov website was searched using the terms heat shock protein, HSPA, Hsp70, Grp78 or mortalin and cancer.

## 3. Results

### 3.1. Similarities and Differences in HSP70 Functions

The HSP70 proteins have similar structural domains and chaperone functions responsible for proper folding of their client proteins [5,12] (Figure 2).

The *N*-terminal contains a nucleotide binding domain responsible for hydrolyzing ATP to generate the energy needed to drive the client protein folding and release functions. The C-terminal region contains a substrate or peptide-binding domain, which binds to exposed hydrophobic areas of newly synthesized polypeptide chains or unfolded/misfolded client proteins. These two domains are connected by a flexible linker domain. Adjacent to the substrate binding domain is a lid domain that can fold over the substrate binding domain during folding reactions. The HSP40 chaperones assists in bringing client proteins to ATP-bound HSP70 chaperones, then triggers the HSP70 ATP hydrolysis function and subsequently leaves HSP70s in the ADP-bound state. Increasing evidence supports that HSP70s only partially fold the client protein, which then spontaneously folds into the proper conformation upon release from the Hsp70 chaperone complex [12]. NEFs preferentially bind to ADP-bound HSP70s and stabilize a conformation of HSP70s that allows release of ADP. Subsequent binding of ATP to the HSP70/NEF complex causes release of the NEF and completion of a cycle that allows the regenerated ATP-bound HSP70 to fold proteins again. Additional proteins become involved in this cyclic process to modulate protein aggregation and degradation. Another activity of HSP70-containing complexes is to pull unfolded proteins through a membrane pore using an entropy-based process, which brings newly synthesized proteins to their primary organelle localizations [10]. The HSP70 chaperones also exert similar effects at the cellular level and in cancer [5]. The main differences in Hsp70, hsc70, Grp78 and mortalin HSP70 chaperone functions (Table 1) are related to their primary locations within or outside of the cell (Table 2) as described in detail below.

#### 3.1.1. Hsp70

The Hsp70-1 and Hsp70-2 proteins encoded by the HSPA1 and HSPA2 genes, respectively, differ by only two amino acids and are difficult to differentiate in experimental assays, and therefore will be referred to in this article collectively as Hsp70, unless the specific analysis or report is focused on the individual Hsp70-1 or Hsp70-2 proteins. Hsp70 is primarily located in the cytoplasm and nucleus, but also is present at the lysosome and in membrane-anchored and secreted states, especially in diseased conditions. Cellular and extracellular levels of Hsp70 are increased by heat and other stresses [15]. Hsp70 is the only one of the Hsp70, hsc70, Grp78 and mortalin group that has been shown to be induced by heat. Heat shock increases the Hsp70 content of exosomes, while having no effect on the rate of exosome secretion [16]. Hsp70 cellular and physiologic functions in health and cancer have been extensively reviewed elsewhere [13] and are highlighted in contrast to hsc70, Grp78 and mortalin in Figure 3 and described below.

#### 3.1.2. hsc70

The hsc70 protein differs in amino acid sequence from Hsp70 by only 25%, and also differs in that it is constitutively expressed. The primary localization of hsc70 is in the cytoplasm where it has been shown to support multiple oncogenic processes through its regulation of client proteins. For example, Hsc70 binds and folds newly synthesized cyclin D1 and supports assembly and function of the cyclin D1/cyclin dependent kinase 4 (CDK4) holoenzyme [17]. Cyclin D1 is amplified and overexpressed in multiple cancers and the cyclin D1/CDK4/6 holoenzyme complex is stimulated by mitogenic signaling cascades to accelerate cancer cell proliferation [18]. In colon cancer cells, Hsc70 prevented proteomic stress-induced degradation of the Ras family member Rab1A and apoptosis [19]. Hsc70 has been studied most extensively in glioma where its regulation of proliferation and apoptosis was shown to involve binding and regulation of β4-galactosyltransferase 5 [20]. Overall, depletion of hsc70 significantly reduces cell proliferation, migration and invasion, and promotes cell apoptosis in cancer cells [21,22].

Roles of hsc70 in maintaining cellular homeostasis involve its ability to shuttle proteins between the cytoplasm and organelles. In transporting proteins between the cytoplasm and lysosomes, hsc70 is the only chaperone known to mediate a cellular recycling program called chaperone mediated autophagy (CMA). To perform CMA, hsc70 binds unfolded/misfolded proteins that have an exposed KFERQ amino acid sequence motif and brings them to the lysosome [23]. The hsc70/KFERQ-containing client protein complex then interacts and binds to the cytosolic tail of the monomer form of lysosome-associated membrane protein 2A (LAMP-2A), which acts as a receptor in CMA. A conformational change of LAMP-2A then takes place and the client protein is transferred to a lysosomal form of Hsc70 (lys-Hsc70), which pulls the client protein into the lysosome for degradation. Subsequently, LAMP-2A is disassembled into its monomer form to bind to new proteins [24,25,26]. Through the degradation of soluble cytosolic substrates, CMA plays a role in the regulation of the cell cycle and diseases. For example, in many types of cancer cells CMA is upregulated and is considered a pro-survival pathway. In this instance, CMA plays a protective role in stress-induced microenvironments and nutrient depletion [26]. However, in non-cancer cells CMA has anti-tumorigenic functions [24,26]. Two recent reviews explain in greater detail the role of CMA in diseases [24,26]. The hsc70 chaperone also mediates an endosomal-selective form of microautophagy (EmiA), in which hsc70 loads KFERQ-containing proteins into late endosomes/multivesicular bodies that eventually fuse into the lysosome for degradation of the contents [27,28]. Hsp70 is also present at the lysosome where it protects against lysosomal permeabilization and subsequent cell death [29]. Hsc70 appears to regulate shuttling of various proteins, including MHC major histocompatibility complex class II (MHC II), into exosomes [30].

Another unique property of hsc70 is that it shuttles between the cytoplasm and nucleus while facilitating import/export of different client proteins into the nucleus. Specifically, hsc70 controls nucleocytoplasmic transport systems by facilitating nuclear export of import receptor proteins, such as importin α/β and transportin [31]. Under stress, hsc70 is retained in the nucleus thereby limiting its function to the nuclear compartment [32]. Upon recovery from stress, hsc70 is released from nucleus and normal nucleocytoplasmic transport is re-established [32]. Lack of hsc70 relocation into the cytoplasm has been shown to alter the ability of cells to survive under stress conditions [33]. In esophageal cancer cells, hsc70 has been found to be upregulated and localized to nuclear pore complex [34].

Although hsc70 is constitutively expressed within the cell, several studies have demonstrated that its plasma membrane and secreted forms can be stimulated by factors that do not affect membrane or secreted Hsp70. Secretion of hsc70 protein is induced by contact inhibition or serum deprivation and repressed by the cathepsin D lysosomal protease, while intracellular levels remain the same [35]. This secreted hsc70 inhibited proliferation and stimulated cellular contact inhibition. Use of peptide sequencing and specific antibodies demonstrated that the secreted 70kD protein consisted entirely of hsc70, and not hsp70. Hsp70 and hsc70 differ in their inducibility by cytokines. Interferon γ (IFNγ) induced intracellular Hsp70, but not hsc70, while increasing plasma membrane levels of hsc70, but not Hsp70 [36]. The Hsp70 and hsc70 proteins are similar in that both are released from cells during viral infection [37]. Extracellular Hsp70 and hsc70 are similar in that both have a dual effect on the tumor immune microenvironment, by stimulating and suppressing anti-tumor immune responses [38,39,40].

#### 3.1.3. Grp78

Grp78 is primarily localized in the endoplasmic reticulum (ER), however, it has been observed also in the cytosol, nucleus, mitochondria and plasma membrane, and extracellular space [41]. Grp78 is also present in exosomes. Exosomal levels of Grp78 are reduced by histone de-acetylase inhibitors, which lead to increased levels of acetylated Grp78 that become aggregated in the ER and bound by VPS34, a class III phosphoinositide-3 kinase, which mediates autophagic degradation of the Grp78 aggregates [42]. Grp78 is induced by stresses, such as accumulation of unfolded proteins in the ER, but not by the heat shock. At the ER membrane under non-stressed conditions, Grp78 binds and inhibits, inositol-requiring transmembrane kinase/endoribonuclease 1α (IRE1α), PKR-like ER kinase (PERK) and Activating Transcription Factor 6 (ATF6). Upon buildup of unfolded/misfolded proteins in the ER, Grp78 releases IRE1α, PERK and ATF6, which allows them to induce the unfolded protein response (UPR) [43]. UPR restores ER homeostasis and supports cell survival by increasing the ratio of chaperone to general protein synthesis and directing retrograde transport of misfolded proteins back through the ER to the cytoplasm for degradation by the ubiquitin-proteasome system (UPS). Once the buildup of unfolded/misfolded proteins is reduced, normal Grp78 expression and binding to IRE1α, PERK and ATF6 are restored [43]. In situations where UPR cannot restore homeostasis, uncontrolled UPR leads to macroautophagy or apoptosis [44]. Macroautophagy, also called autophagy, can be induced by a variety of stimuli, such as nutrient deprivation or Grp78 upregulation, and involves formation of autophagosomes surrounding cytoplasmic areas, including organelles, that fuse with lysosomes to degrade the contents and release the components for recycling [43]. Uncontrolled UPR leads to apoptosis through upregulation of CCAAT/-enhancer-binding protein homologous protein (CHOP), which induces expression of Growth arrest and DNA damage 34 (GADD34), unless autophagy induction is sufficient to prevent this [43].

Grp78 has been shown to be upregulated in various cancers, which contributes to increased tumor cell proliferation and stemness, and angiogenesis, invasion, metastasis and chemotherapy resistance through its endogenous cytoprotective mechanisms and altered metabolism [45,46,47,48,49,50]. Grp78 suppresses immune responses through a variety of mechanisms including alterations in lipid metabolism [51]. Overexpression of Grp78 has been also shown to exert anti-apoptotic function by stabilizing mitochondrial permeability, reducing the release of mitochondrial cytochrome c and suppressing caspase-7/12 activation [52]. High Grp78 levels have been associated with more aggressive features and worse prognosis [53]. Grp78 expressed at the surface of cancer and endothelial cells, regulates various signaling cascades which are pro-proliferative, antiapoptotic and promigratory, which has been reviewed elsewhere [54]. Extracellular Grp78 was shown to inhibit immune responses to tumor metastases in the liver [30] and tumor response to the anti-angiogenic agent, Bortezomib [55].

#### 3.1.4. Mortalin

After translation in the cytoplasm, mortalin is transported primarily into mitochondria, but also has been observed in the cytosol, nucleus, ER, cytoplasmic vesicles, plasma membrane and extracellular space [56,57]. Mortalin is not induced by heat but can be induced by ionizing radiation and oxygen or glucose deprivation [58,59,60]. This chaperone has been shown to regulate diverse cellular functions, including proliferation, stress response, chromosome replication and apoptosis [61]. Its activity and function are determined by localization in the cell and binding partners. In the mitochondria, mortalin forms the core essential ATPase component of the import machinery for mitochondrial proteins that are encoded by the nuclear genome and synthesized in the cytoplasm [62]. Thus, mortalin function is essential for maintenance of the mitochondrial integrity, energy metabolism, free-radical generation and biogenesis [61]. Furthermore, mortalin is upregulated by a mitochondrial version of UPR (UPRmt) and assists in UPRmt alleviation of unfolded proteins in the mitochondria [63,64]. Just outside the mitochondria, mortalin couples the inositol 1, 4, 5-triphosphate receptor (IP3R) on ER to the voltage dependent anion channel (VDAC1) on mitochondria to facilitate Ca^2 +^ transfer from the ER lumen to the mitochondrial matrix [65,66,67]. In the nucleus, mortalin facilitates maintenance of telomere length, and also regulates genetic processes through control of centrosome duplication during chromosome replication and division, and mRNA processing and transport [57]. Mortalin has been shown to play a role in cellular release of exosomes [68,69], while Hsp70 present on the surface of cells and exosomes can stimulate an immune reaction and lead to tumor immunity and autoimmunity [70,71]. Although secreted mortalin does not appear to regulate the immune function, mortalin is stimulated by complement attack to relocate from the mitochondria to the plasma membrane where its ATPase binding domain binds C5b-9 complex and prevents it from forming pores that would lyse and kill the cell [72,73,74,75]. Hsp90 also binds C9 and co-operates with hsp70 in protecting cells from complement-mediated cell death [76].

In stressed and cancer cells, cytosolic mortalin sequesters upregulated p53 protein in cytoplasm thereby inhibiting nuclear or mitochondrial p53-induction of apoptosis inducing genes [77,78]. Mortalin also inhibits p53 transcriptional activity when these two proteins are localized in the nucleus [57], and also binds and super-activates the kinase Mps1, which promotes chromosome duplication [79]. Elevated mortalin levels repress p53 at the centrosome and allow aberrant centrosome duplication and consequently survival of cancer cells that are aneuploid as a result of genetic instability or paclitaxel treatment [80].

Overexpression of mortalin is involved in development, progression, metastases and drug resistance of cancers [81,82,83,84]. The mortalin HSPA9 gene was cloned via its differential cytoplasmic staining pattern in immortalized compared to mortal cells [85,86]. Expression of mortalin in cancer at higher levels than in stem cells, including human embryonic stem cells (hESCs) and induced pluripotent stem cells (iPSCs), implicates maintenance of telomere length as a mechanism by which mortalin supports carcinogenesis [87]. Mortalin expression has been shown to promote carcinogenesis, and to be sequentially upregulated with increasing cancer aggressiveness [78,88,89]. The molecular mechanisms of mortalin in driving cancer include upregulation of the MAPK/ERK signaling pathway [82]. Secretion and binding of mortalin and podoplanin has been shown to be coordinated in oral squamous carcinoma cells, while their binding on the cell surface was identified primarily at the invading front of tumors suggesting a pro-migration role for extracellular mortalin [88]. Hsp70 has also been shown to promote cancer cell migration [89,90,91].

### 3.2. Bioinformatic Analysis of Protein Binder Profiles

Cellular effects of Hsp70, hsc70, Grp78 and mortalin are mediated through their effects on client proteins and modulated by other protein binders involved in their function. A proteomic study of client proteins for Hsp70 and hsc70 in human cells identified that they have unique, but overlapping, profiles of co-chaperones and client proteins which change upon presence of unfolded proteins [92]. In this current study, a bioinformatic approach using publicly available data was used to compare the profiles of proteins which directly bind Hsp70-1, Hsp70-2, hsc70, Grp78 and mortalin (Appendix A). The hsc70 chaperone had by far the greatest number of direct binding proteins identified (1352), followed by Grp78 (969), Hsp70 (571) and mortalin (493). Numbers of proteins that uniquely bind to these chaperones or that bind to each set of two of the chaperones were determined (Appendix A and Figure 4).

Bioinformatic analysis confirmed considerable overlap in the canonical pathways, molecular, cellular, physiological and disease functions of the direct binding protein profiles for Hsp70-1, hsc70, Grp78 and mortalin. Comparison of the −Log *p*-values of the canonical pathways associated with the binding protein profiles identified considerable overlap (Figure 5).

As expected, major outliers for individual HSP70s included Huntington’s Disease Signaling for hsc70 [93] (−Log *p* = 33.3), ER Stress Pathway for Grp78 (−Log *p* = 8.6) and Mitochondrial Dysfunction for mortalin (−Log *p* = 10.3). Differences in outliers of pathway significance levels were noted for EIF2 signaling with hsc70 having higher significance EIF2 signaling (−Log = 42.3) compared to Hsp70 (−Log *p* = 6.9), NRF2-mediated Oxidative Stress Response having higher significance in Hsp70 (−Log *p* = 29.8) compared to mortalin (−Log *p* = 8.4), TNFR2 Signaling having higher significance for mortalin (−Log *p* = 23.8) compared to Hsp70 (−Log *p* = 4.0, Regulation of eIF4 and p70S6K Signaling having higher significance for hsc70 (−Log *p* = 21.1) compared to Grp78 (−Log *p* = 3.5) and mTOR signaling with hsc70 having a higher significance (−Log *p* = 17.7) compared to mortalin (−Log *p* = 2.4).

When each HSPA binding profile was evaluated individually, IPA identified cell death and survival and cell cycle as the top two molecular and cellular functions significantly associated with direct protein binding profiles of these chaperones (Appendix A). Additionally, gene expression, cellular development, and cellular growth and proliferation, were included in the top five for each. Gene expression was included in the top five functions associated with Hsp70-1, Hsp70-2, hsc70 and mortalin, while this was replaced with protein synthesis for Grp78. IPA identification of physiological systems significantly associated with the direct protein binding profiles (Appendix A) included organismal survival, embryonic development, connective tissue development and function, and tissue morphology to be in the top five most significantly associated with each of Hsp70-1, hsc70 and Grp78. While mortalin also had embryonic development, organismal survival and tissue morphology in its top five most significantly associated diseases, the other two (lymphoid tissue structure and development and hematological system development and function) in the top five were unique from the other three chaperones evaluated. IPA identified cancer as the disease most significantly associated with each of the of chaperone direct binding protein profiles (Appendix A). Additionally, organismal injury and abnormalities was included as the second most significantly associated disease with hsc70, Grp78 and mortalin and the third diseases most significantly associated with Hsp70-1. Endocrine system disorders were in the top five diseases most significantly associated each of the HSPA chaperones. There were no common individual diseases among the top five significantly associated diseases for these four chaperones.

### 3.3. Utilization as Cancer Biomarkers

In general, the presence of heat shock proteins in the blood has been considered a biomarker of damage, stress or inflammation. High cellular or circulating levels of Hsp70 or Grp78 have been reported to be prognostic in multiple cancers and reviewed elsewhere [94,95,96]. Twenty cancer-focused clinical trials listed on clinicaltrials.gov measure Hsp70 as biomarker of stress, drug response or toxicity, or cancer diagnosis or burden (Appendix A). High cellular or circulating mortalin levels have been reported to be significantly associated with worse outcomes of patients with hepatocellular carcinoma, serous ovarian carcinoma, colorectal cancer, pancreatic cancer, non-small cell lung cancer and invasive ductal carcinoma of the breast [84,97,98,99,100,101,102,103]. There are relatively fewer publications evaluating hsc70 as a prognostic indicator in cancer patients. In a study of clear cell renal cell carcinoma, the presence versus absence of hsc70 expression was significantly associated with worse overall survival [104]. Although expression levels of hsc70 or LAMP2A as indicators of CMA were not associated with each other in pulmonary squamous cell carcinomas, high expression levels of either of these proteins were predictive of worse overall survival [105]. One study of hsc70 as a biomarker (NST03252717: Predictive Role of New Biomarkers for Hypersensitive Patients to Radiation in Breast Cancer (BIORISE)) is listed in clincialtrials.gov. This currently active clinical study aims to validate hsc70 and other identified protein levels in blood as biomarkers of radiation-induced late effects in breast cancer patients. Hsp70 cross-reactivity of the antibodies used in these two hsc70 studies was not reported.

As Hsp70 and hsc70 antibodies are known to have slight cross-reactivity due to the high homology of these two proteins and there are scant reports of hsc70 as a cancer biomarker, this study probed TCGA data to identify significant associations of tumoral hsc70 mRNA expression with survival probability of several cancers. High hsc70 expression was associated with worse survival probability in breast (*p* = 0.019), cervical (*p* = 0.01), hepatocellular carcinoma (*p* = 0.0023) and mesothelioma (*p* = 0.043). In contrast, low hsc70 expression was associated with worse survival probability in renal cell carcinoma (*p* = 0.014). Hsc70 was not associated with survival probability of the other TCGA studied cancers. A TCGA analysis published by others found high hsc70 mRNA expression to be significantly associated with overall survival of acute myeloid leukemia (AML) patients [106].

Hsc70 exhibited gender-specific associations with several cancers in TCGA data. In colorectal cancer, low/medium hsc70 expression predicted worse survival probability compared to high hsc70 expression in males (*p* = 0.027), but not in females (*p* = 0.99). While hsc70 was not associated with overall head and neck squamous carcinoma (HNSCC) survival probability (*p* = 0.23), female patients with high hsc70 expression had significantly worse survival probability compared to females with low hsc70 expression (*p* < 0.0001), males with high hsc70 expression (*p* < 0.0001) and males with low/medium hsc70 expression (*p* = 0.0009), suggesting that hsc70 is prognostic only in females. The worse survival probability of hepatocellular carcinoma cancer patients described in the above paragraph appears to be driven primarily by male patients, since high hsc70 was associated with worse survival probability (*p* < 0.0001), while there was no association of hsc70 expression with survival probability in female patients (*p* = 0.59).

### 3.4. Utilization as Cancer Therapeutic Drug Targets

Cancer therapeutic drugs based on synthetic and natural compounds and biologics are in development for heat shock proteins [107,108]. The compounds being developed for HSP70s in general often bind to more than one of the highly homologous chaperones and are limited in their clinical development due to adverse toxicities [108]. Efforts to improve the therapeutic ratios (toxicity/efficacy) of HSP70 targeted drugs have included designing the investigational agents to bind allosteric pockets outside of the nucleotide and peptide binding domains of the chaperone molecules, or to be selective for individual HSPs and/or subcellular compartments [108,109]. Currently the only small molecule HSP70 inhibitor in clinical trial as a cancer therapeutic is SHetA2, which is being evaluated in an oral capsule formulation for treatment of advanced or recurrent gynecologic cancers (NCT 04928508).

Promising immune-based approaches have taken advantage of the immune stimulatory property of Hsp70 to stimulate an anti-tumor immune response [110,111]. There are six cancer therapeutic clinical trials utilizing Hsp70 vaccines alone or in combination with drugs or other antigens listed on clinicaltrials.gov (Table 3), however, the results of these studies have not yet been reported. One of these trials (NCT00005633) is a Phase 1 trial comparing the tolerability and immune response of three doses of Hsp70 and Grp78 antigens combined with other immunogens in advanced stage melanoma. There is one therapeutic trial listed for a Grp78 inhibitor on clinicaltrials.gov. This Phase 1 study found that a monoclonal antibody to Grp78 (PAT-SM6) was well tolerated and was associated with 33.3% stable disease in relapsed or refractory multiple myeloma [112].

Cancer therapeutic strategies to avoid development of drug resistance combine drugs that target complementary intracellular signaling pathways. The goal is for one drug to prevent cancer cells from developing alterations in one pathway that can bypass the inhibition of the complementary pathway targeted by the other drug. Preclinical studies support combinations of Hsp70 inhibitors with Hsp90 inhibitors or cisplatin [113]. Current evidence supports inhibiting mortalin to prevent or overcome cancer resistance to inhibitors of the raf/MAPK/MEK/ERK signaling pathway [114,115] and to paclitaxel [116] and to complement inhibition of mutant p53 [78]. Clinical studies indicate that Grp78 upregulation is a cause and prognostic biomarker of cancer resistance to chemotherapy, while preclinical and clinical studies provide data to support the theory that inhibition of Grp78 can increase chemotherapy and molecularly targeted agents with acceptable toxicity [96].

### 3.5. Utilization in Cancer Prevention Strategies

Vaccination against Hsp70 has been studied in clinical trials for prevention of high- risk breast and cervical cancers. The clinicaltrials.gov website lists three trials of vaccine therapy using DNA plasmids expressing Hsp70 for prevention of cancer in patients with pre-neoplastic conditions (Table 4). All of these studies evaluate prevention of cervical cancer in women diagnosed with (CIN) lesions, atypical squamous cells (ASC) or low-grade squamous intraepithelial lesion (LSIL). One of these trials (NCT00121173) has reported results, which show that three out of nine (33%) patients receiving the highest of three doses of the vaccine had no cervical intraepithelial lesions (CINs) assessed by colposcopy and biopsy(ies) at the week 15 visit. There were no serious adverse events or mortality in the study.

The role of mortalin in cellular immortalization and early carcinogenesis support its targeting in cancer prevention studies. A major consideration in development of cancer chemoprevention strategies is to avoid unnecessary toxicity to the patients. The selective presence of mortalin chaperone complexes with client proteins such as p53 in cancer compared to healthy cells [77] indicates that the drugs targeted at disruption of mortalin/client protein complexes will not have significant toxicity. Indeed, minimal to no toxicity has been proven for drugs and a natural product shown to disrupt the mortalin/p53 complex [117,118,119]. Renal toxicity which halted clinical development of the MKT-077 allosteric inhibitor of mortalin [120,121] appears to be caused by nonspecific accumulation of the drug in kidneys [122]. Disruption of mortalin/p53 complexes by SHetA2 was shown to prevent establishment of ovarian cancer without toxicity in a preclinical model of ovarian cancer maintenance therapy [78], thus suggesting a role for mortalin inhibitors in secondary cancer prevention. In this preclinical study, synergistic activity in cell culture and additive activity in the animal model was observed between SHetA2 and PRIMA-1^MET^, a drug that reactivates wild type p53 apoptosis activity in mutant p53 molecules. Lack of significant toxicity of SHetA2 at doses 50-fold above the effective dose in animal models, suggests high potential for this agent in cancer chemoprevention [117].

## 4. Discussion

HSP70 maintenance of proper subcellular localization, folding and function of a wide range of client proteins and complexes provides a survival mechanism to protect cells experiencing stress. The Hsp70, hsc70, Grp78 and mortalin HSP70 family members have overlapping and unique activities that are likely needed to fine-tune cell survival responses and maintenance of homeostasis. Differences in their inducibility, subcellular localization, co-chaperones and affinity for binding unique profiles of client proteins provide the cell with a way to calibrate survival responses to stress. For each of the chaperones, cell death and survival, and cell cycle were identified as the top two cellular processes, and cancer was identified as the top disease most significantly affected by their unique profile of direct binding proteins. Pre-cancer and cancer cells that upregulate these chaperones develop an evolutionary advantage that drives tumorigenesis and cancer therapy resistance. In general, cancer patients with higher levels of these HSP70 proteins have worse prognosis.

The upregulation of HSP70 proteins in cancer provides an opportunity to image cancers based on heightened cell surface expression, and to develop drugs with selective cytotoxicity to cancer cells without harming healthy cells. Despite promising pre-clinical evidence for multiple HSP70 natural and synthetic small molecule inhibitors, only SHetA2 is currently being tested in a cancer therapeutic clinical trial. Targeting mortalin with MKT-077 failed in clinical trials due to non-specific toxicity. Strategies that target Hsp70, hsc70, Grp78 and mortalin in development of cancer therapeutics are limited by the high homology of these proteins making it difficult to take advantage of the selective properties of the individual chaperones. As has been suggested for Hsp90 inhibitors [4], targeting the compounds to protein domains responsible for specific protein–protein interactions offers a reasonable approach for optimizing HSP70-based therapy. Therefore, there is a critical need for development of experimental tools that can differentiate the individual proteins and their specific interactions with co-chaperones and client proteins.

In this study, HSP70-1, hsc70, Grp78 and mortalin each were found to directly bind hundreds of proteins in profiles that partially overlapped. A logical approach to HSP70 drug discovery would be to first target direct-protein binders for the individual HSP70s, and then consider combining various drugs with complementary targets. This type of precision medicine approach focusing on specific molecular alterations in the cancer target tissue may also need to take into consideration the type of cancer targeted. For example, in this study we found that hsc70 overexpression has significant prognostic value in breast cancer, cervical cancer and mesothelioma, and gender-specific association in HNSCC and hepatocellular carcinoma, while low hsc70 expression was significantly associated with worse prognosis in renal cell carcinoma. These types of selective associations of individual HSP70 proteins with prognosis of different cancers may be due to specific molecular alterations that commonly occur in different cancers but may also be affected by more complex anatomical and physiological factors.

Other important considerations for developing anti-cancer strategies targeted at the HSP70 proteins is co-operation of various HSP types with the HSP70s in protein folding and cellular compensation to HSP70 inhibition by upregulation of other heat shock proteins. For instance, treatment of ovarian cancer cells with an Hsp90 inhibitor led to increased expression of Hsp70, hsc70, Hsp27 and Hsp47 [123]. On the other hand, genetic reduction of hsc70 expression led to increased expression of Hsp70, and no effect on Hsp90 inhibitor sensitivity in multiple colon and ovarian cell lines [124]. However, when both Hsp70 and hsc70 were simultaneously inhibited, there was an increase in proteasomal degradation of Hsp90 client proteins, cell cycle arrest and apoptosis. The complementary effects of inhibiting HSP70s and Hsp90 suggests that combinations of drugs targeted at different HSP proteins could be used to optimize and personalize cancer therapies, once sufficient information can be gathered about the milieu of chaperones, co-chaperones and client proteins in a patient’s cancer. Additionally, drugs, such as SHetA2, that simultaneously target multiple HSP70 proteins could be used. A promising aspect of these studies was that simultaneous inhibition of Hsp70, hsc70 and Hsp90, or hsc70, Grp78 and mortalin, induced apoptosis in cancer cells with drastically reduced effects in non-cancer cells.

Preclinical data support targeting cell surface HSP70 proteins to stimulate anti-cancer immune responses, image tumors and alter cancer-driving signaling down-stream of the cell surface HSP70s [125,126]. Multiple trials have been conducted using Hsp70 as an antigen to stimulate immune attack against developing or established tumors in cancer prevention and therapy, respectively. Grp78 is also beginning to be targeted in these strategies. Given the known immune regulation by Hsp70, hsc70, Grp78 and mortalin, they all represent rational targets for development of immune-based cancer prevention and treatment strategies that prime immune cells with the HSP70 antigens.

A unique strength of this review is the systematic delineation of different direct binding proteins and their signaling pathways for Hsp70, hsc70, Grp78 and mortalin. Very limited studies have differentiated HSP70 proteins on the basis of their specific client proteins or binding partners. In this quest, for the first time, we provided a detailed bioinformatic analysis showing overlapping and unique binding partners associated with overlapping and outstanding pathways for the different HSP70s. A weakness of this study is that it is limited to only four HSP70s, while the other HSP70 proteins not discussed are likely to influence the activities of Hsp70, hsc70, Grp78 and mortalin.

## 5. Conclusions

The protection of pre-cancer and cancer cells by elevated HSP70 proteins represents a rational target for imaging, diagnostic and drug development strategies. The complexity of their multiple cellular and extracellular locations, co-chaperones and client proteins offers an opportunity to fine-tune and personalize the strategies for individual cancers or tumors.

## Figures and Tables

**Figure 1 cells-10-02996-f001:**
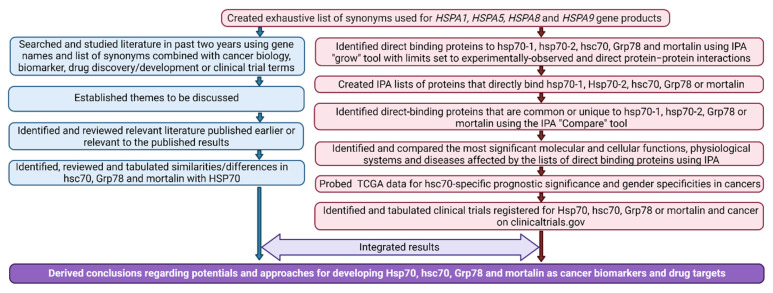
Methodologic approach.

**Figure 2 cells-10-02996-f002:**
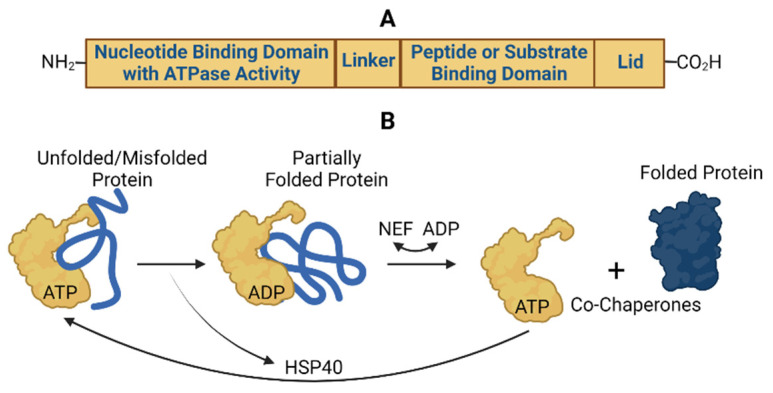
Common structural elements (**A**) and protein folding functions (**B**) of Hsp70, hsc70, Grp78 and mortalin.

**Figure 3 cells-10-02996-f003:**
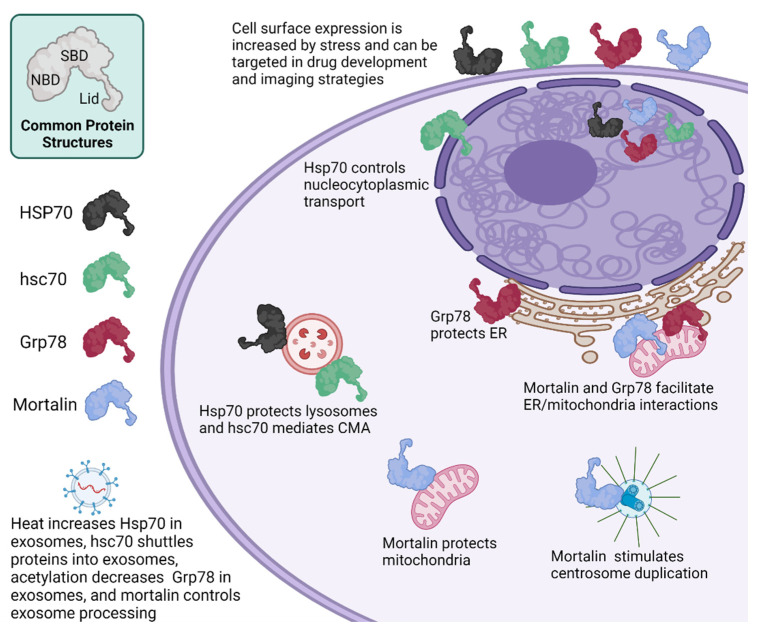
Major areas of cell protection by various HSP70 proteins.

**Figure 4 cells-10-02996-f004:**
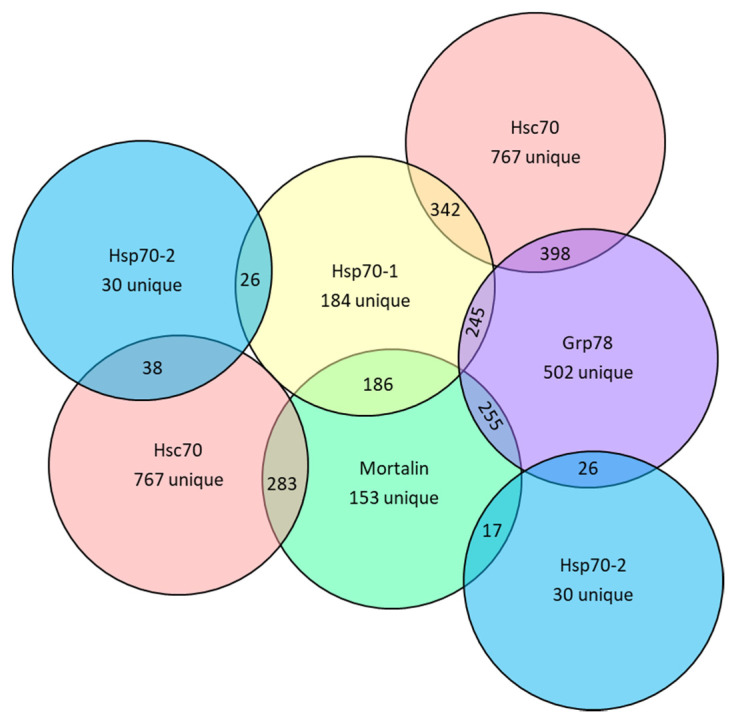
Comparison of protein binding partners for Hsp70, hsc70, Grp78 and mortalin. The numbers inside the non-overlapping, or overlapping, areas of the circles represent the number of direct binding proteins that are unique to the indicated HSPA chaperone, or are common to the adjacent HSPA chaperones, respectively.

**Figure 5 cells-10-02996-f005:**
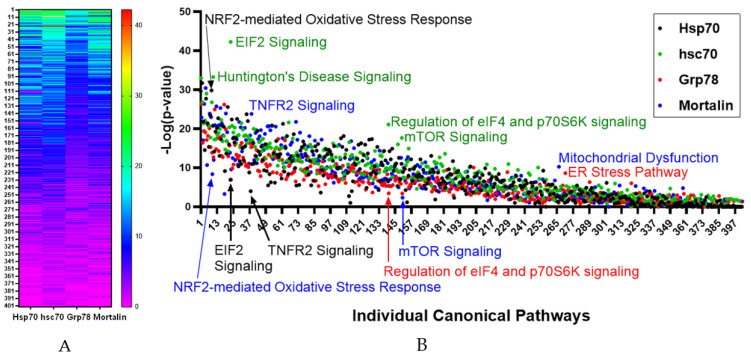
Comparison of significant canonical pathways associated with protein binding profiles of different HSP70s. (**A**) Heat map of −Log *p* values for pathways presented on the *y*-axis. (**B**) Plot of −Log *p* values for each pathway and HSP70 protein with discussed-outliers labeled.

**Table 1 cells-10-02996-t001:** Comparison of HSP70 protein functions.

HSP70 Protein	ATPase	Protein Folding	Facilitates Protein Degradation Through:	Protein Localization Control	Lysosome Protection	Immune Function Regulation	Cell Cycle, Survival and Death
Hsp70	X	X	UPS		X	X	X
Hsc70	X	X	CMA, EmiA	Cytoplasm vs. lysosome or nucleus		X	X
Grp78	X	X	UPR, UPS, macroautophagy	Retrograde ER transport		X	X
Mortalin	X	X	UPRmt	Cytoplasm vs. mitochondria or nucleus		X	X

**Table 2 cells-10-02996-t002:** Comparison of cellular localizations of HSP70 proteins.

	Cytoplasm	Nucleus	Mitochondria	ER	Plasma Membrane	Extracellular	Exosomes
Hsp70	++	+			+	+	+
Hsc70	++	T			+	+	+
Grp78	+	+	+	++	+	+	+
Mortalin	+	+	++	+	+	+	+

+: observed to be present at indicated location; ++: primarily present at indicated location; T: transiently present at indicated location.

**Table 3 cells-10-02996-t003:** Therapeutic clinical trials testing Hsp70 and/or Grp78 (BiP) antigens.

Title	Status	Intervention	ID
Study Using Vaccination with Heat Shock Protein 70 (HSP70) for the Treatment of CML in Chronic Phase	Completed	Heat Shock Protein 70 HSP70	NCT00027144
Vaccine Therapy in Treating Patients with Chronic Myelogenous Leukemia	Completed	Recombinant 70-kD heat-shock protein; Ganetespib; Sirolimus	NCT00030303
Vaccine Therapy in Treating Patients with Stage III or Stage IV Melanoma	Completed	OVA BiP peptide; gp209-2M antigen; recombinant 70-kD heat-shock protein; tyrosinase peptide	NCT00005633
AG-858 in Patients Who Are Cytogenetically Positive After Treatment with Gleevec™	Terminated	Autologous HSP-70 Protein-Peptide Complex (AG-858) Plus Gleevec™	NCT00058747
Targeted Natural Killer (NK) Cell Based Adoptive Immunotherapy for the Treatment of Patients with Non-Small Cell Lung Cancer (NSCLC) After Radiochemotherapy (RCT)	Suspended	Hsp70-peptide TKD/IL-2 activated, autologous NK cells	NCT02118415
Personalized Cancer Vaccine in Egyptian Cancer Patients	Recruiting	Peptide cancer vaccine	NCT05059821
Vaccine Therapy in Treating Patients with Stage III or Stage IV Melanoma	Completed	Anti-GRP78 monoclonal IgM antibody PAT-SM6	NCT01727778

**Table 4 cells-10-02996-t004:** Hsp70 vaccination cervical cancer prevention trials.

Title	Status	Intervention	ID
Vaccine Therapy with or Without Imiquimod in Treating Patients With Grade 3 CIN	Recruiting	TA-HPV, pNGVL4a-Sig/E7(detox)/HSP70 DNA vaccine, imiquimod	NCT00788164
Vaccine Therapy in Preventing Cervical Cancer in Patients with CIN	Completed	pNGVL4a-Sig/E7(detox)/HSP70 DNA vaccine	NCT00121173
Phase II Study of Treatment for HPV16+ ASC-US, ASC-H and LSIL	Recruiting	PVX-2; placebo	NCT03911076

## Data Availability

Data is provided in Appendix A.

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
