# Peer review of "Similarities and Differences of Hsp70, hsc70, Grp78 and Mortalin as Cancer Biomarkers and Drug Targets"

_cells, 2021, doi:10.3390/cells10112996_

Round 1

Reviewer 1 Report

The manuscript by Rajani Rai et al shows in-depth analysis and definitely has scope to gain the attention of readers.I would like to share some serious concerns with authors:

1) A graphical abstract should be constructed for this article so that the concept is understandable by a non-bioinformatics background personnel.

2) Lots of statistical and system biology related work was done in this article, but I did not find any network topology, confidence score of networks, pathway enrichment pie charts/bar diagrams. Since lots of information has been extracted and well documented, a graphical representation is must.

3) The methodology should be demonstrated by flow diagram.

4) Without any pictorial presentation of data it looks very complex and difficult to understand and soon readers will lose interest.

5) I highly recommend that the authors depict the extracted and screened data in the form of graphical representations. I am sure this upgrade will uplift the quality and gravity of this article.

6) Discussion section should be highly improved and well elaborated. 

Author Response

We thank the reviewer for helpful comments, which we have incorporated into the revision as described below:

The manuscript by Rajani Rai et al shows in-depth analysis and definitely has scope to gain the attention of readers.I would like to share some serious concerns with authors:

1) A graphical abstract should be constructed for this article so that the concept is understandable by a non-bioinformatics background personnel.

A graphical and a video abstract are provided.

2) Lots of statistical and system biology related work was done in this article, but I did not find any network topology, confidence score of networks, pathway enrichment pie charts/bar diagrams. Since lots of information has been extracted and well documented, a graphical representation is must.

Graphical representations are provided in Figures 2-5.

3) The methodology should be demonstrated by flow diagram.

A methodology flow diagram is provided in Figure 1.

4) Without any pictorial presentation of data it looks very complex and difficult to understand and soon readers will lose interest.

Graphical representations are provided in the Graphical Abstract and Figures 2 - 5.

5) I highly recommend that the authors depict the extracted and screened data in the form of graphical representations. I am sure this upgrade will uplift the quality and gravity of this article.

Graphical representations are provided in the Graphical Abstract and Figures 2 -5.

6) Discussion section should be highly improved and well elaborated. 

The Discussion section was extensively improved and elaborated and a separate Conclusion section was added.

Reviewer 2 Report

Rai et al., has submitted the review entitled “Similarities and differences of Hsp70, hsc70, Grp78 and Mortalin as cancer biomarkers and drug targets”. Overall the review covers the broader content and emphasizes essential aspects of cancer. However, this reviewer has some suggestions to improve the manuscript quality further.

  1. The authors should illustrate the protein structures and their differences (e.g., hsc70 see https://doi.org/10.3390/cells8080849, https://doi.org/10.1038/s41573-019-0036-1).
  2. Also, a table that describes the different functions of proteins should be added.
  3. The entire manuscript should be spell-checked to avoid mistakes, e.g., line 109 “comples”.

Author Response

We thank the reviewer for helpful comments, which we have incorporated into the revision as described below:

Rai et al., has submitted the review entitled “Similarities and differences of Hsp70, hsc70, Grp78 and Mortalin as cancer biomarkers and drug targets”. Overall the review covers the broader content and emphasizes essential aspects of cancer. However, this reviewer has some suggestions to improve the manuscript quality further. 

  1. The authors should illustrate the protein structures and their differences (e.g., hsc70 see https://doi.org/10.3390/cells8080849, https://doi.org/10.1038/s41573-019-0036-1).

Figures 2 and 3 and the graphical abstract provides illustrations similar to those recommended.

  1. Also, a table that describes the different functions of proteins should be added.

Tables 1 and 2 have been added to the manuscript to describe the protein functions and their cellular localizations.

  1. The entire manuscript should be spell-checked to avoid mistakes, e.g., line 109 “comples”.

We apologize for the typos and have corrected them.

Reviewer 3 Report

The manuscripts “Similarities and Differences of Hsp70, hsc70, Grp78 and Mortalin as Cancer Biomarkers and Drug Targets” by Rai et al summarizes current literature about the heat shock proteins of the HSP70 family. They explain similarities and distinct differences between the different HSP70 isoforms (mortalin, GRP78 and hsc70) and try to relate it with cancer, thereby focusing on function as biomarker, as drug target or as cancer prevention strategy.

It is an interesting summary about the different HSP70 isoforms, but not that helpful for interested readers if they want to learn more about HSP70 inhibitors as the topic is more scratched on the surface and not described in detail. Which inhibitors exist in particular for the isoforms that were focused on? In which entities were the inhibitors tested? Did any inhibitors reach to clinical trials, if yes, what were therapeutic side-effects?

e.g. In terms of biomarkers, GRP78 is described as potential biomarker for Multiple Myeloma, a malignant plasma cell disease.

Minor:

Typos:

(Supplemntary Table 5) line 321

Author Response

We thank the reviewer for helpful comments, which we have incorporated into the revision as described below:

The manuscripts “Similarities and Differences of Hsp70, hsc70, Grp78 and Mortalin as Cancer Biomarkers and Drug Targets” by Rai et al summarizes current literature about the heat shock proteins of the HSP70 family. They explain similarities and distinct differences between the different HSP70 isoforms (mortalin, GRP78 and hsc70) and try to relate it with cancer, thereby focusing on function as biomarker, as drug target or as cancer prevention strategy.

It is an interesting summary about the different HSP70 isoforms, but not that helpful for interested readers if they want to learn more about HSP70 inhibitors as the topic is more scratched on the surface and not described in detail. Which inhibitors exist in particular for the isoforms that were focused on? In which entities were the inhibitors tested? Did any inhibitors reach to clinical trials, if yes, what were therapeutic side-effects?

The preclinical and clinical trial results and discussion of HSP70 isoform drugs have been substantially expanded and listed in Tables 4, 5 and S6.

e.g. In terms of biomarkers, GRP78 is described as potential biomarker for Multiple Myeloma, a malignant plasma cell disease.

Grp78 as a cancer biomarker is mentioned in the second sentence of section 3.1 Utilization as Cancer Biomarkers. Also, use of an antibody against GRP78 in a clinical trial of Multiple Myeloma is now described and cited in reference 114 and listed in Table 3.

Minor:

Typos:

(Supplemntary Table 5) line 321

We apologize for the typos and have corrected them.

Round 2

Reviewer 1 Report

All raised issues has been resolved by the authors.

Reviewer 2 Report

The revised version of the manuscript is acceptable for publication. Hereby I endorse the manuscript for publication. 

Reviewer 3 Report

The manuscript has substantially been improved with the included information and can be published now.